

# Interplay between chiral media and perfect electromagnetic conductor plates: Repulsive vs. attractive Casimir force transitions

Thomas Oosthuyse[1,2★] and David Dudal[1‡]

**1** KU Leuven Campus Kortrijk – Kulak, Department of Physics,
Etienne Sabbelaan 53 bus 7657, 8500 Kortrijk, Belgium
**2** Ghent University, Department of Physics and Astronomy,
Krijgslaan 281-S9, 9000 Gent, Belgium

★ david.dudal@kuleuven.be , † thomas.oosthuyse@kuleuven.be

## Abstract

We determine the Casimir energies and forces in a variety of potentially experimentally viable setups, consisting of parallel plates made of perfect electromagnetic conductors (PEMCs), which generalize perfect electric conductors (PECs) and perfect magnetic conductors (PMCs), and Weyl semimetals (WSMs). Where comparison is possible, our results agree with the Casimir forces calculated elsewhere in the literature, albeit with different methods. We find a multitude of known but also new cases where repulsive Casimir forces are in principle possible, but restricting the setup to PECs combined with the aforementioned WSM geometry, results in purely attractive Casimir forces.


# 1 Motivation

It is well known by now that according to the no-go theorem [1], the Casimir force is always attractive between dielectrics which are related by reflection. However, when this reflection symmetry is absent there are no clear rules regarding if the Casimir force is attractive or repulsive, and it has to be considered on a case by case basis. Therefore it is interesting to look at the well known, and geometrically symmetric, parallel plate setup, and study in what cases repulsive Casimir forces are possible.

The case of parallel dielectric plates is well known, see e.g [2]. Perhaps surprisingly, no repulsive Casimir force is possible in vacuum, even when the plates have differing dielectric functions $\epsilon_1 \neq \epsilon_2$. To permit a repulsive Casimir force one has to fill the empty space between the plates with a third dielectric medium and have $\epsilon_1 > \epsilon_3 > \epsilon_2$ where $\epsilon_3$ is the dielectric function of the internal region. This has been experimentally verified with gold, silica, and liquid bromobenzene [3].

Another way to circumvent the no-go theorem in a geometrically symmetric setup, is to move away from dielectric materials. Having one plate magnetically permeable, approximated by a perfect magnetic conductor (PMC) instead of a perfect electric conductor (PEC), already results in a repulsive Casimir force [4,5], as well as interesting behavior under the zero plate separation limit [6]. PMCs can possibly be realized using metamaterials [7]. This can be generalized towards perfect electromagnetic conductors (PEMCs) [8–13]. Other materials of interest are chiral materials, both for the plates and for filling the gap between them [10,14–20]. Weyl semimetals specifically are of interest to us, as they have a convenient description in terms of a quantum field theory effective action [21,22].

We focus on a case similar to [10,18,23], where a WSM-like medium is placed between perfect electric conductors (PECs) or PEMCs. The resulting Casimir force is attractive at short and long ranges with a repulsive region at intermediate separation. The catch is that in those setups the WSM is assumed to change in size to always completely fill the gap, i.e. it has to be fluid or gas-like, something which does not appear very realistic. Therefore in this paper we build up towards the case of a slab of WSM with fixed width placed between PEMCs and see if the features of [10,18] survive.

# 2 Overview of calculation method

We first give an overview of the calculation method developed in [10,24]. All setups in this paper will be derived from the Euclidean QED action, augmented with a classical but space-time dependent $\theta$-term

$$S = \int \mathrm{d}^4 x \left[ \frac{1}{4} F_{\mu\nu} F_{\mu\nu} + i\theta(x) \frac{1}{4} F_{\mu\nu} \widetilde{F}_{\mu\nu} \right], \tag{1}$$

where $\widetilde{F}_{\mu\nu} = \frac{1}{2} \varepsilon_{\mu\nu\rho\sigma} F_{\rho\sigma}$ is the dual field strength tensor and $\theta(x)$ is a background axion-like field [25]. This action has the property that it can model a variety of different physical situations by choosing the background field $\theta(x)$, see e.g. [21,22,26]. In practice our background field will only depend on $x^3 \equiv z$ such that translation invariance is preserved in the $t, x$ and $y$ directions. The equations of motion only depend on the gradient of $\theta(z)$, therefore we will denote it by $\beta(z) \equiv \partial_z \theta$ such that $\beta(z) = 0$ corresponds to conventional QED in which case $\theta$ becomes an irrelevant constant. In principle we could allow discontinuities in $\theta(z)$, which would correspond to surfaces on which Hall currents are possible, but we leave this for a future work.

We add a Feynman gauge fixing term

$$S_{\text{gf}} = \int d^4x\, \frac{1}{2}(\partial A)^2\,, \tag{2}$$

to the action. Feynman gauge is chosen because it allows us to diagonalize the Lorentz structure of the propagators notwithstanding that the translation invariance in the $z$-direction is broken due to the $\beta(z)$ background. Overall, this gauge choice does not influence physics, but it does simplify the computations.

We will consider three types of medium with which to build our setups. The first one is the trivial case of the QED vacuum, with $\beta(z) = 0$. The second type of media are Weyl semimetals (WSMs), which can be modelled by setting $\beta(z)$ equal to a nonzero constant [21, 22]. In this case the constant value of $\beta(z)$ can be interpreted as the separation of the two Weyl nodes of the WSM in momentum space [27]. The last type of media are perfect electromagnetic conductors (PEMC), which are modelled by applying the boundary conditions

$$n_\mu F_{\mu\nu} + i \cot\alpha\, n_\mu \widetilde{F}_{\mu\nu} \bigg|_\Sigma = 0\,, \tag{3}$$

on the surface $\Sigma$, with normal vector $n_\mu$ [8–10]. These boundary conditions have the feature that they are the most general linear boundary conditions that are still compatible with gauge invariance [28]. PEMC materials are a generalization of perfect electric conductors (PEC) and perfect magnetic conductors (PMC), where the parameter $\alpha \in \left[-\frac{\pi}{2}, \frac{\pi}{2}\right]$ can be used to interpolate between the PEC ($\alpha = 0$) and PMC ($\alpha = \pm\frac{\pi}{2}$) boundary conditions.

The relevant boundary conditions can be included into the action by introducing a lagrange multiplier field $b_\mu^a$ in a term

$$S_{\text{bc}} = \int_\Sigma d^3\mathbf{x}\, b_\mu^a G_\mu^a(A)\,, \tag{4}$$

where the index $a$ includes all surfaces on which boundary conditions need to be applied, and $G_\mu^a(A) \equiv n_\rho F_{\rho\mu} + i \cot\alpha_a n_\rho \widetilde{F}_{\rho\mu}$ encodes the boundary conditions. The boundary conditions in our case will only be applied to static plates perpendicular to the $z$-axis, so $n_\mu = \delta_{\mu 3}$.

Given the total action $S_{\text{tot}} = S + S_{\text{gf}} + S_{\text{bc}}$ we can now perform a redefinition of the fields to 'complete the square' and bring it into the form

$$S_{\text{tot}} = \frac{1}{2} \int \frac{d^3\mathbf{k}}{(2\pi)^3} \int dz\, A_\mu^\dagger K_{\mu\nu} A_\nu + \frac{1}{2} \int \frac{d^3\mathbf{k}}{(2\pi)^3} b_i^{\dagger a} \mathbb{K}_{ij}^{ab} b_j^b\,, \tag{5}$$

where we Fourier transformed all coordinates except the $z$-coordinate and suppressed the $\mathbf{k}$ dependence. The kinetic operator is given by

$$K_{\mu\nu} = \delta_{\mu\nu}(\partial_z^2 - |\mathbf{k}|^2) + \beta(z)\varepsilon_{\mu\nu i3}k_i\,, \tag{6}$$

or, in the polarization basis (see Appendix B) where the Lorentz structure is diagonal

$$K_{rs} = \text{diag}(\partial_z^2 - |\mathbf{k}|^2,\quad \partial_z^2 - k_c^2(z),\quad \partial_z^2 - (k_c^\star)^2(z),\quad \partial_z^2 - |\mathbf{k}|^2)\,, \tag{7}$$

and we have defined $k_c^2(z) = |\mathbf{k}|^2 + i|\mathbf{k}|\beta(z)$. The $A_\mu$ field can be integrated out, after which the boundary conditions encoding $b^a$ fields form a non-local 3D effective theory with kinetic operator

$$\mathbb{K}_{ij}^{ab} = \overline{V}_{i\mu}^a(\partial_z) V_{\nu j}^b(\partial_{z'}) K_{\mu\nu}^{-1}(z, z') \bigg|_{z=z_a, z'=z_b}\,, \tag{8}$$

where $z_a$ stands for the $z$-coordinate of the $a$-th plate, and $K^{-1}_{\mu\nu}(z,z')$ is the Green's function of the $K_{\mu\nu}$ operator. The matrices $\overline{V}^a_{i\mu}, V^a_{\nu j}$ follow from the boundary conditions, and are given in the polarization basis as

$$
\begin{aligned}
V^a_{rs}(\partial_z) &= \begin{pmatrix} \partial_z & 0 & 0 & i|\mathbf{k}| \\ 0 & \partial_z + i\cot\alpha_a|\mathbf{k}| & 0 & 0 \\ 0 & 0 & \partial_z - i\cot\alpha_a|\mathbf{k}| & 0 \end{pmatrix}, \\
\overline{V}^a_{rs}(\partial_{z'}) &= \begin{pmatrix} \partial_{z'} & 0 & 0 \\ 0 & \partial_{z'} + i\cot\alpha_a|\mathbf{k}| & 0 \\ 0 & 0 & \partial_{z'} - i\cot\alpha_a|\mathbf{k}| \\ -i|\mathbf{k}| & 0 & 0 \end{pmatrix}.
\end{aligned}
\tag{9}
$$

The (unregularized) Casimir energy per unit surface area of the system follows as $\mathcal{E} = \mathcal{E}_A + \mathcal{E}_b$, with

$$
\begin{aligned}
T\mathcal{V}_2\mathcal{E}_A &= \frac{1}{2}\log\det(K), \\
T\mathcal{V}_2\mathcal{E}_b &= \frac{1}{2}\log\det(\mathbb{K}),
\end{aligned}
\tag{10}
$$

where $T\mathcal{V}_2$ is the 3D spacetime volume of the transversal space.[1] $\mathcal{E}_A$ is not present in the conventional Casimir effect between two parallel PEC plates as being trivial. The Casimir energy density $\mathcal{E}_b$ resulting from the boundary conditions can be calculated directly as

$$
\mathcal{E}_b = \frac{1}{2}\int\frac{\mathrm{d}^3\mathbf{k}}{(2\pi)^2}\log(|\mathbb{K}|),
\tag{11}
$$

where $|\mathbb{K}|$ is the matrix determinant of the $\mu, \nu, a, b$ indices of $\mathbb{K}^{ab}_{\mu\nu}$. In the case of two parallel plates this expression can be regularized (renormalized) by subtracting $\mathcal{E}_b$ in the limit of infinite plate separation. The Casimir energy density $\mathcal{E}_A$ is more difficult to calculate however due to the broken translation symmetry in the $z$-direction. Instead it is easier to calculate the Casimir force $F_A = -\frac{\mathrm{d}\mathcal{E}_A}{\mathrm{d}L}$ directly, where $L$ is the relevant separation parameter between media and/or plates. Using Jacobi's formula the Casimir force follows as

$$
F_A = -\frac{1}{2}\int\frac{\mathrm{d}^3\mathbf{k}}{(2\pi)^3}\int\mathrm{d}z\,\mathrm{tr}\left[\frac{\mathrm{d}K}{\mathrm{d}L}K^{-1}\bigg|_{z=z'}\right],
\tag{12}
$$

where the "tr" stands for the trace over the Lorentz indices. In the case we consider $\beta(z)$ is a piecewise constant function, where the parameter $L$ then varies the size of the intervals where $\beta(z)$ is constant. It follows that the derivative w.r.t. $L$ results in Dirac delta-distributions which make the $z$ integration trivial. Unlike for the conventional Casimir effect, now the Casimir force $F_A$ also needs a proper regularization, which is consistent with the fact that one can construct new vacuum counterterms with $\beta(z)$, or better said, with its dimensional parameters.

Now only the calculation of the Green's function $K^{-1}_{\mu\nu}(z,z')$ is left. This is a difficult problem for general $\beta(z)$, but can be done systematically for piecewise constant $\beta(z)$ by solving the translation invariant equations[2]

$$
K^a_{\mu\rho}D^a_{\rho\nu}(z-z') = \left[\delta_{\mu\rho}\left(\partial_z^2 - |\mathbf{k}|^2\right) + \beta_a\varepsilon_{\mu\rho i}k_i\right]D^a_{\rho\nu}(z-z') = \delta_{\mu\nu}\delta(z-z'),
\tag{13}
$$

where $\beta_a$ are the constant values taken by $\beta(z)$. In the polarization basis this reduces to solving for a single type of Green's function

$$
\left(\partial_z^2 - k^2_{ca}\right)\varphi_a(z-z') = \delta(z-z'),
\tag{14}
$$

---

[1] We implicitly take the limits of infinite timespan $T \to \infty$ and infinite surface area $\mathcal{V}_2 \to \infty$.

[2] No sum over $a$.

where $k_{ca}^2 = |\mathbf{k}|^2 + i\beta_a|\mathbf{k}|$, which has as solution

$$\varphi_a(z - z') = -\frac{1}{2k_{ca}} e^{-|z-z'|k_{ca}}. \tag{15}$$

The full Green's function $K^{-1}(z, z')$ can then be constructed by gluing the individual $D^a(z-z')$ together and demanding smoothness and continuity (for $z \neq z'$) by adding solutions to the homogeneous differential equations. The gluing of the $D^a(z-z')$ would be different in the case that $\theta(z)$ has discontinuities, i.e. Hall currents are present. More details about the calculations in this section can be found in [10].

## 3 The Casimir force in specific setups

We now have the tool set to calculate Casimir forces in setups consisting of WSMs and PEMC materials. We will discuss a few setups which are more experimentally viable than the one discussed in [10] or [18]. Indeed, as concrete experimental verifications of the standard Casimir force for parallel plates all rely at some point on a variable separation between the plates, see e.g. [29, 30], it does not appear possible to ever measure the Casimir force between the two sides of a WSM or alike when the size of that WSM has to be changeable.

To be as general as possible, PEMC materials are used as plates/boundary conditions to explore more exotic behaviors of the Casimir force, such as repulsive forces, but the PEC limit (or even PMC) can always be taken to arrive at more conventional setups.

Two simple, but interesting, cases are the Casimir force between two slabs of WSMs [14, 15, 31] and the Casimir force between a WSM and a PEMC material. The case of the Casimir force between two PEMC materials has already been discussed in [10]. We also discuss how the presence of a slab of WSM of finite width modifies the Casimir force between two PEMC materials, this can be seen as an experimentally viable modification of the more mathematically idealized setups discussed in [10, 18].

We will often compare with the QED Casimir force and energy between parallel PEMC plates [9, 10]

$$\begin{aligned}
\mathcal{E}_{\text{qed}}(L, \alpha) &= -\frac{1}{8\pi^2 L^3} \operatorname{Re} \operatorname{Li}_4(e^{2i\alpha}), \\
F_{\text{qed}}(L, \alpha) &= -\frac{3}{8\pi^2 L^4} \operatorname{Re} \operatorname{Li}_4(e^{2i\alpha}),
\end{aligned} \tag{16}$$

where $\alpha$ is the difference between the duality angles of the PEMC plates $\alpha = \alpha_2 - \alpha_1$, and $\operatorname{Li}_n(x)$ is the polylogarithm. The conventional QED Casimir effect then follows as $\mathcal{E}_{\text{qed}}(L) \equiv \mathcal{E}_{\text{qed}}(L, 0) = -\frac{\pi^2}{720 L^3}$ and $F_{\text{qed}}(L) \equiv F_{\text{qed}}(L, 0) = -\frac{\pi^2}{240 L^4}$.

Our $\beta$ parameter is a rescaled version of the one typically found in the literature $\beta = \frac{e^2}{2\pi^2}\beta'$. In WSMs we typically have $\beta'^{-1} \sim 1\,\text{nm}$, see e.g. [32–34], which corresponds to $\beta^{-1} \sim 200\,\text{nm}$. Values of $\beta L$ of the order of $\beta L \sim 1$ to 20 correspond to distances of the order of $L \sim 0.2\,\mu\text{m}$ to $4\,\mu\text{m}$, which are reachable in experiments [29, 35, 36].

### 3.1 PEMC plate and semi-infinite WSM

Consider a setup where a semi-infinite slab of WSM is separated from a PEMC material by a vacuum gap of width $L$ as seen in FIG. 5 in the appendix. The WSM can be modelled by setting $\beta(z)$ to

$$\beta(z) = \begin{cases} \beta, & \text{if } z < -\dfrac{L}{2}, \\ 0, & \text{if } z > -\dfrac{L}{2}, \end{cases} \tag{17}$$

while the PEMC material follows from the boundary condition part of the action

$$S_{\text{bc}} = \int d^3\mathbf{x}\, b_i n_\mu \big(F_{\mu i} + i\cot\alpha\, \widetilde{F}_{\mu i}\big)\bigg|_{z=\frac{L}{2}}. \tag{18}$$

In this case the Casimir force will arise solely from the $\mathcal{E}_b$ contribution, as $F_A$ results in an $L$-independent infinite constant, to be removed with a vacuum counterterm. The Casimir energy can be regularized by subtracting the $L \to \infty$ limit, and results in

$$\mathcal{E}_1 = \text{Re} \int \frac{d^3\mathbf{k}}{(2\pi)^3} \log\bigg(1 - \frac{k_c - |\mathbf{k}|}{k_c + |\mathbf{k}|} e^{-2|\mathbf{k}|L + 2i\alpha}\bigg), \tag{19}$$

where $k_c^2 = |\mathbf{k}|^2 + i\beta|\mathbf{k}|$. The Casimir force follows as $F_1 = -\frac{d\mathcal{E}_1}{dL}$, and is completely attractive in the PEC limit $\alpha \to 0$, while completely repulsive in the PMC limit $\alpha \to \pm\frac{\pi}{2}$. We can study the Casimir force relative to the QED Casimir force $\widetilde{F}_1(\beta L, \alpha) = \frac{F(\beta, L, \alpha)}{F_{\text{qed}}(L)}$ which is dimensionless. This expression can be expanded to find that the long range $\beta L \gg 1$ force tends to the QED Casimir force

$$\widetilde{F}_1(\beta L, \alpha) = \widetilde{F}_{\text{qed}}(\alpha) - \frac{1575}{16\pi^{\frac{7}{2}}\sqrt{\beta L}} \text{Re}\big[(1-i)\text{Li}_{\frac{7}{2}}\big(e^{2i\alpha}\big)\big] + \frac{360}{\pi^4 \beta L} \text{Im}\,\text{Li}_3\big(e^{2i\alpha}\big) + \mathcal{O}(\beta L)^{-2}, \tag{20}$$

where $\widetilde{F}_{\text{qed}}(\alpha) \equiv \frac{F_{\text{qed}}(L, \alpha)}{F_{\text{qed}}(L)} = \frac{90}{\pi^4} \text{Re}\,\text{Li}_4\big(e^{2i\alpha}\big)$. For short ranges $\beta L \ll 1$ the Casimir force follows an $L^{-2}$ power law in the case of a PEC or PMC plate ($\alpha = 0, \pm\frac{\pi}{2}$), instead of the $L^{-4}$ power law for perfectly conducting plates. These are special cases however, and it follows an $L^{-3}$ power law for general $\alpha$. $\widetilde{F}_1(\beta L, \alpha)$ is shown in FIG. 1 up to $\beta L = 20$.

## 3.2 PEMC plate and WSM with finite width

We can also consider the slab of WSM to have a finite width $d$. This can be incorporated by setting $\beta(z) = 0$ for $z < -\frac{L}{2} - d$. The resulting regularized Casimir energy follows as

$$\mathcal{E}_2 = \text{Re} \int \frac{d^3\mathbf{k}}{(2\pi)^3} \log\big(1 - R e^{-2|\mathbf{k}|L + 2i\alpha}\big), \tag{21}$$

with

$$R = \frac{2(k_c + |\mathbf{k}|)(k_c - |\mathbf{k}|)\sinh(k_c d)}{(k_c + |\mathbf{k}|)^2 e^{k_c d} - (k_c - |\mathbf{k}|)^2 e^{-k_c d}}, \tag{22}$$

which returns to (19) in the limit $d \to \infty$. We can once again study the Casimir force relative to the QED Casimir force $\widetilde{F}_2(\beta L, \beta d, \alpha) = \frac{F_2(L, \beta, d, \alpha)}{F_{\text{qed}}(L)}$. The long range $\beta L \gg 1$ behavior is modified compared to the infinite width case:

$$\widetilde{F}_2(\beta L, \beta d, \alpha) = \frac{90}{\pi^4} \text{Re}\bigg[\text{Li}_4\bigg(\frac{e^{2i\alpha}\beta d}{\beta d - 2i}\bigg)\bigg(1 - \frac{4}{3}\frac{(\beta d - 3i)\beta d}{(\beta d - 2i)\beta L}\bigg) + \mathcal{O}(\beta L)^{-2}\bigg], \tag{23}$$

where we see that the $\alpha$ dependency approaches its $\beta L \to \infty$ behavior faster than in the infinite width case. Importantly this also modifies the zero-points of the Casimir force, which can be observed in FIG. 2, where $\widetilde{F}_2(\beta L, \beta d, \alpha)$ is shown for $\beta d = 2$.

As a final remark note that the Casimir energy in this case obeys a similar sum rule as in [9], $\int_{-\frac{\pi}{2}}^{\frac{\pi}{2}} \mathcal{E}_2(L, \beta, d, \alpha)\, d\alpha = 0$ as it follows from [37, 4.225(4)]. A fortiori, the Casimir force obeys the same sum rule.

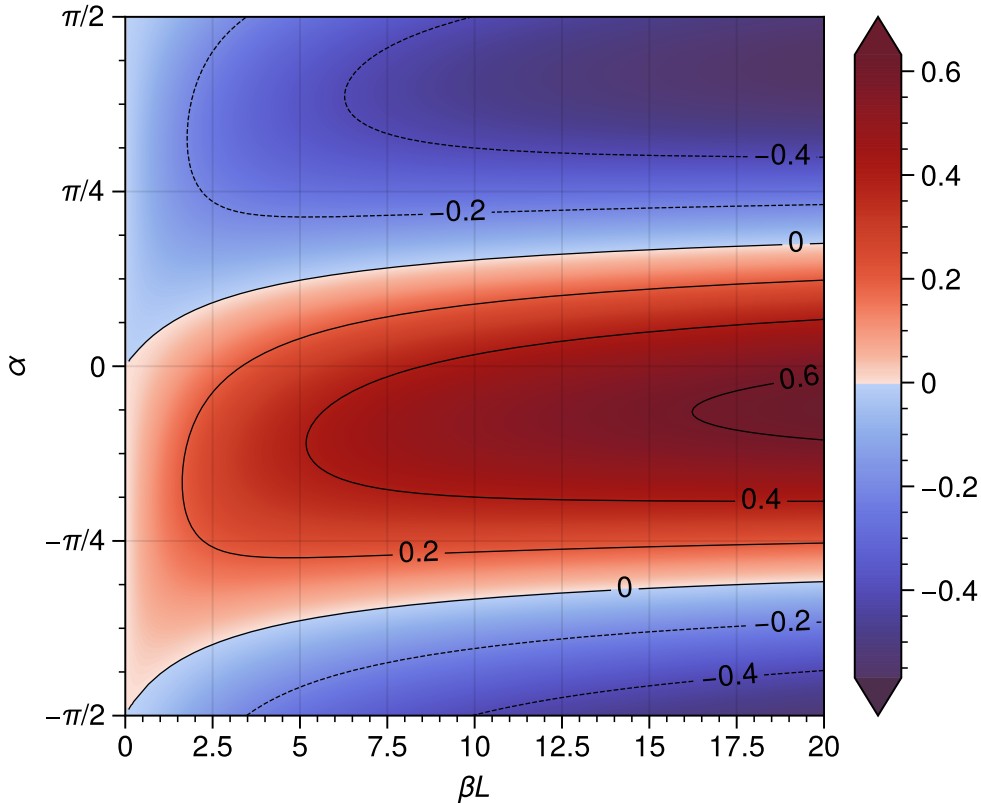

Figure 1: The Casimir force between a PEMC and a semi-infinite slab of WSM, relative to the QED Casimir force $\widetilde{F}_1(\beta L, \alpha) = \frac{F_1(L,\beta,\alpha)}{F_{\mathrm{qed}}(L)}$. Attractive and repulsive forces have been shaded red and blue respectively.

### 3.3 Semi-infinite slabs of WSMs

We now take a look at the Casimir force between two semi-infinite slabs of WSMs separated by a distance $L$, see FIG. 6. We model this by

$$
\beta(z) = \begin{cases} \beta_1, & \text{if} \quad z < -\dfrac{L}{2}, \\[2mm] 0, & \text{if} \quad z \in \left[-\dfrac{L}{2}, \dfrac{L}{2}\right], \\[2mm] \beta_2, & \text{if} \quad z > \dfrac{L}{2}, \end{cases} \tag{24}
$$

where $\beta_1$ is the material parameter of the leftmost WSM, and $\beta_2$ that of the rightmost WSM. As there are no PEMC materials present in this setup, there is no need for auxiliary fields or extra boundary conditions. Consequently only the propagator needs to be calculated and (12) can be used directly. The regularized Casimir force follows as

$$
\begin{aligned}
F_3 = -\beta_1 \beta_2 \operatorname{Re} \int \frac{\mathrm{d}^3 \mathbf{k}}{(2\pi)^3} |\mathbf{k}| e^{-|\mathbf{k}|L} (|\mathbf{k}| + k_{c1})^{-1} (|\mathbf{k}| + k_{c2})^{-1} \\
\times \left[ |\mathbf{k}|(k_{c1} + k_{c2}) \cosh(|\mathbf{k}|L)(|\mathbf{k}|^2 + k_{c1}k_{c2}) \sinh(|\mathbf{k}|L) \right]^{-1},
\end{aligned} \tag{25}
$$

which is invariant under $\beta_1 \leftrightarrow \beta_2$ and $\beta_\alpha \to -\beta_\alpha$. The Casimir force relative to the QED one, $\widetilde{F}_3(\beta_1 L, \beta_2 L) = \frac{F_3(L,\beta_1,\beta_2)}{F_{\mathrm{qed}}(L)}$, is shown in FIG. 3, in terms of the parameters $b_\pm = \frac{1}{2}(\beta_1 \pm \beta_2) \geq 0$,

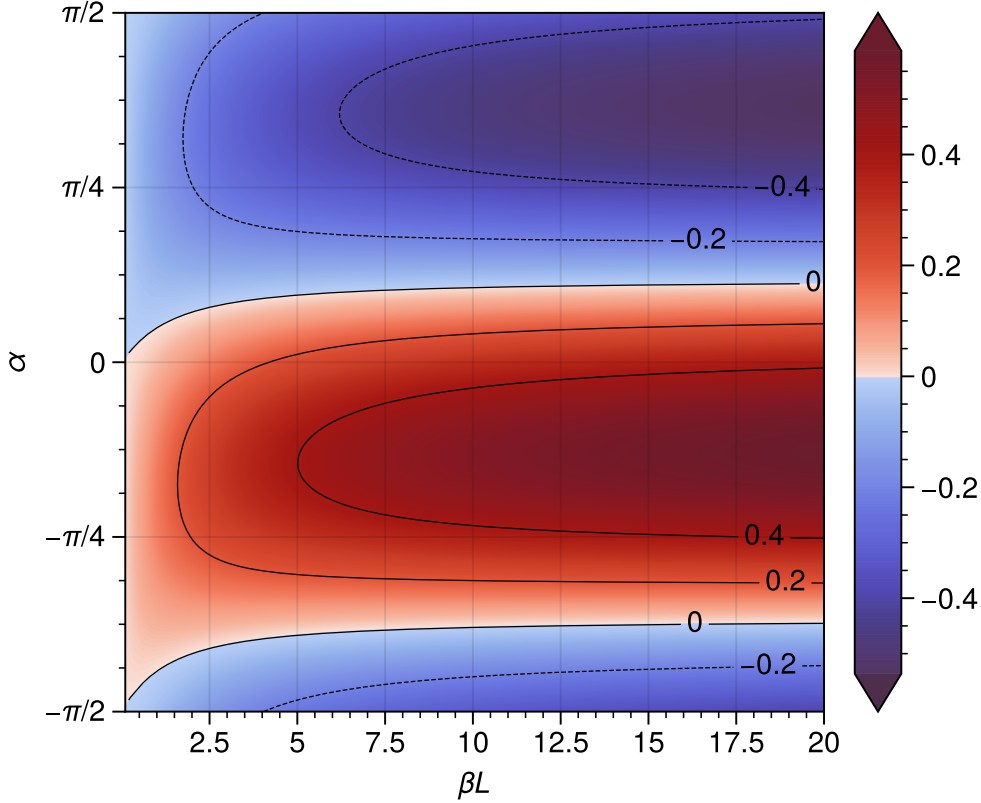

Figure 2: The Casimir force between PEMC and a slab of WSM with width $d$, relative to the QED Casimir force $\widetilde{F}_2(\beta L, \beta d, \alpha) = \frac{F_2(L, \beta, d, \alpha)}{F_{\text{qed}}(L)}$. Attractive and repulsive forces have been shaded red and blue respectively ($\beta d = 2$).

as any set of $\beta_\alpha$ can be mapped to positive $b_\pm$ using the aforementioned symmetries of (25). Interestingly the Casimir force is exclusively attractive in the region $b_- > b_+$, i.e. when $\beta_1$ and $\beta_2$ have opposite sign. When $b_- < b_+$ the Casimir force is repulsive at short while attractive at long distances. Therefore in this situation there exists a stable separation $L$ where the Casimir force is zero. The results of this section agree with [14, 15].

For the sake of completeness, this expression can be integrated over $L$ to obtain the Casimir energy. There is the freedom of subtracting an $L$-independent constant from said energy, which can be used to set the energy at $L \to \infty$ to zero, effectively regularizing the Casimir energy in the usual way. The regularized Casimir energy density is given by $\mathcal{E}_3$ in Appendix C.

### 3.4 A finite width slab of WSM between parallel PEMC plates

At last, let us have a look at what we consider to be a more experimentally viable option of the setups in [10, 18]: a WSM of finite width $d$ confined between two PEMC plates located at $z = z_a$, with $a = 1, 2$, where $z_1$ is the leftmost and $z_2$ the rightmost plate, shown in FIG. 7. Placing the center of the WSM at $z = 0$ such that $z = \pm\frac{d}{2}$ are the boundaries of the WSM, we can use the offsets of the plates to the WSM $z_- = z_1 + \frac{d}{2}$ and $z_+ = z_2 - \frac{d}{2}$ to simplify expressions. There is vacuum between the plates and the WSM slab.

With some effort, the same techniques as for the previous cases can be used to calculate the full Casimir energy $\mathcal{E}_4$, which is given in Appendix C.

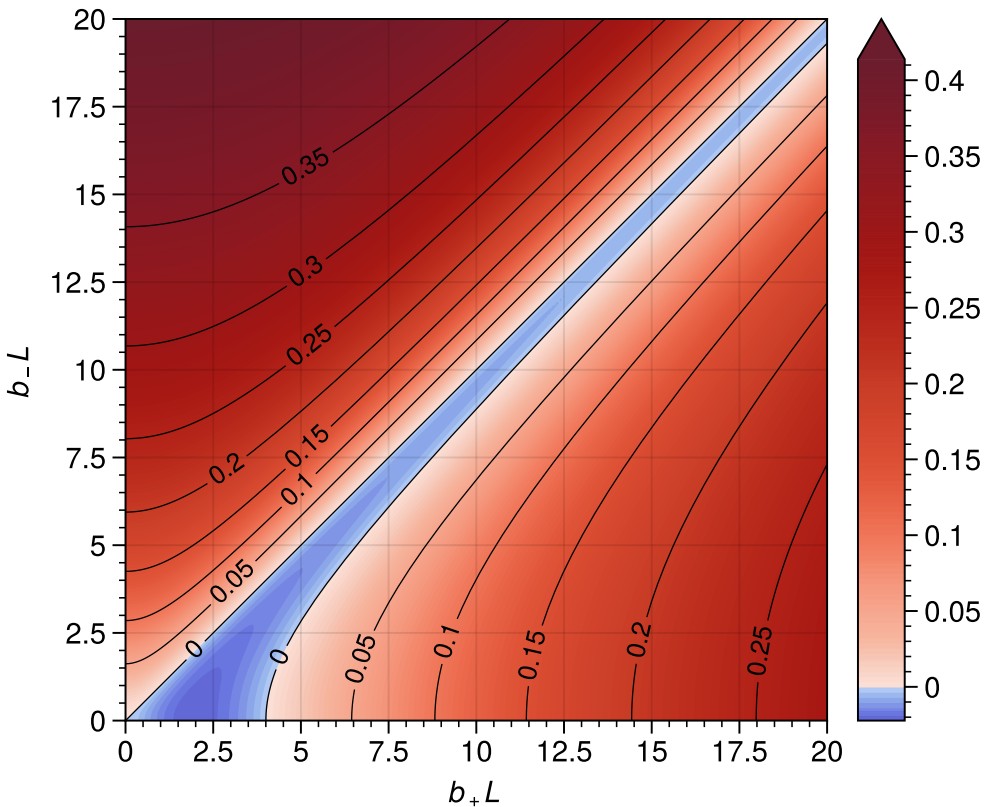

Figure 3: The Casimir force between two semi-infinite WSMs, separated by a distance $L$, relative to the QED Casimir force $\widetilde{F}_3(\beta_1 L, \beta_2 L) = \frac{F_3(L,\beta_1,\beta_2)}{F_{\text{qed}}(L)}$, with $b_\pm = \frac{1}{2}(\beta_1 \pm \beta_2)$. The repulsive region has been highlighted in blue.

As a consistency check we note that in the limit $\beta d \to \infty$, where we keep $z_\pm$ constant, corresponding to an infinitely thick WSM, the Casimir energy reduces to

$$\lim_{\beta d \to \infty} \mathcal{E}_4(z_+, z_-, \beta, d, \alpha_1, \alpha_2) = \mathcal{E}_1(-z_-, \beta, -\alpha_1) + \mathcal{E}_1(z_+, \beta, \alpha_2), \tag{26}$$

i.e. the sum of the Casimir energy of two independent PEMC–WSM setups. Similarly the Casimir energy becomes the PEMC–PEMC Casimir energy in the limit $\beta d \to 0$

$$\mathcal{E}_4(z_-, z_+, \beta, d, \alpha_1, \alpha_2) \overset{\beta d = 0}{=} \mathcal{E}_{\text{qed}}(z_+ - z_- + d, \alpha_2 - \alpha_1). \tag{27}$$

More interestingly, the Casimir force acting on the rightmost plate reads $F_4 = -\frac{d\mathcal{E}_4}{dz_+}$. The case where the leftmost plate is a PEC and attached to the WSM ($z_- = 0$) is shown in FIG. 4, relative to the QED Casimir force $\widetilde{F}_4(\beta z_+, \beta d, \alpha_2) = \frac{F_4(0,z_+,\beta,d,0,\alpha_2)}{F_{\text{qed}}(L)}$, for $\beta d = 2$. It can be seen that the short range interaction is comparable to the case without the leftmost PEC as in FIG. 2, but for longer range the PEC–PEMC interaction starts to dominate. In other words a thin film of WSM applied to an electric conductor could be used to modify the short range behavior of the Casimir force. A general observation is that the force $F_4$ remains purely attractive for all $\beta, d, z_\pm$ in the case of PEC plates. Similarly the Casimir force is purely repulsive for all combinations of parameters when the plates are PMC. As the PEMC–PEMC and PEMC–WSM (both infinite and finite width) Casimir forces are also attractive (repulsive) for PEC (PMC) plates, this is perhaps no unexpected result. This implies however that if one replaces the idealized (unrealistic) setup of [10, 18] by the more realistic case of a finite slab of WSM

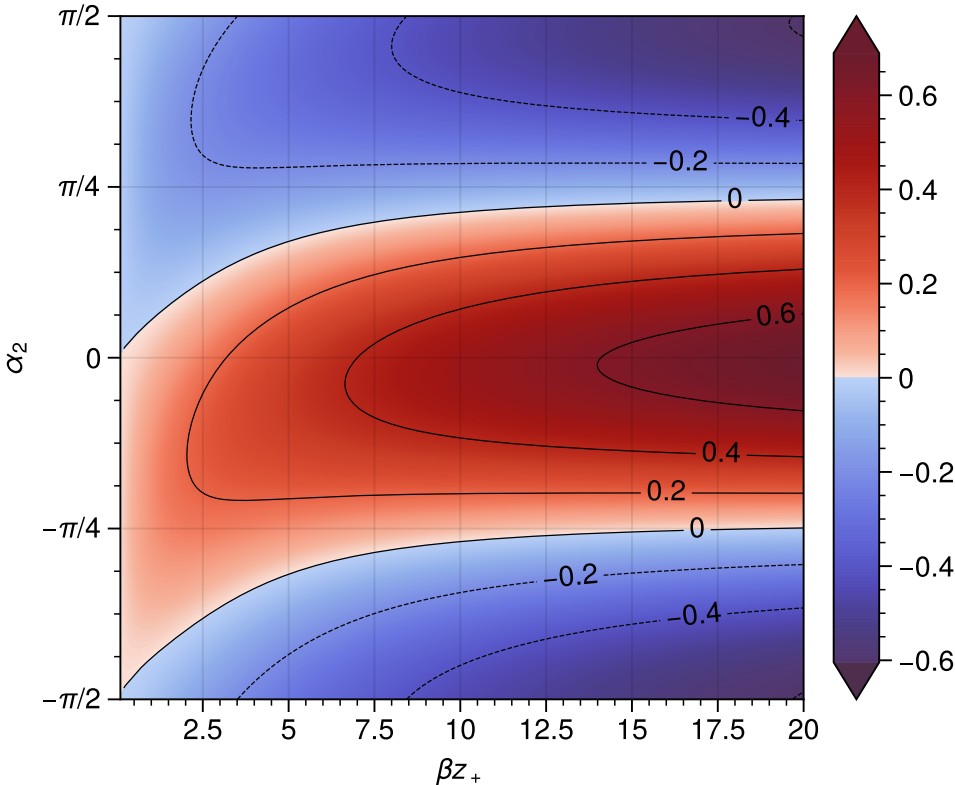

Figure 4: The Casimir force acting on the rightmost PEMC plate in a PEC–WSM–PEMC setup, where the PEC is attached to the WSM ($z_- = 0$), relative to the QED Casimir force $\widetilde{F}_4(\beta z_+, \beta d, \alpha_2) = \frac{F_4(0, z_+, \beta, d, 0, \alpha_2)}{F_{\text{qed}}(z_+)}$. Attractive and repulsive forces have been shaded red and blue respectively ($\beta d = 2$).

between PEC plates, there is actually no repulsive Casimir force present anymore, which was one of the reasons to consider chiral media such as WSMs in Casimir-like setups to begin with.

Our finding has yet another consequence, namely that if the distance between the plates is kept constant and we look at the force acting on the WSM, it can be seen that the WSM placed at the center between the plates is an unstable (stable) stationary point for PEC (PMC) plates.

Noteworthy, for general $\alpha_a \in \left[-\frac{\pi}{2}, \frac{\pi}{2}\right]$, the behavior of the system is much richer, allowing for the presence of repulsive-attractive and attractive-repulsive transitions as seen in FIG. 4, much like the WSM–PEMC configurations in FIG. 1 and FIG. 2. This makes clear the potential huge relevance of using PEMC materials for the plates around a WSM to hopefully study in the future their "Casimir phase diagram" in an experimental setting.

## 4 Conclusions

We have calculated the Casimir force in a multitude of configurations consisting of PEMCs and WSMs by making use of the path integral formalism. We found a variety of situations in which a repulsive Casimir force is possible, although one is required to either have a general PEMC, i.e. not a PEC, present or two WSMs with $\beta_i$ of the same sign.

Using general PEMCs it should be possible to tune the Casimir force to the desired behavior, being either completely attractive, repulsive, or a mixture of both depending on the plate separation.

As PEMCs can be seen as a special case of bi-isotropic materials with infinite material parameters [8, 12, 13], it could be interesting to consider the Casimir effect between bi-isotropic materials as the most realistic scenario to generalize and test our predictions. A covariant description of such materials, or even simpler dielectric materials, following [38] might perhaps shed new light on efficient calculation methods of the Casimir effect in exotic yet realistic materials.

## Acknowledgments

We thank F. Canfora, P. Pais, L. Rosa and S. Stouten for useful discussions.

**Funding information**   The work of D.D. and T.O. was supported by KU Leuven IF project C14/21/087.

## A   The considered geometries

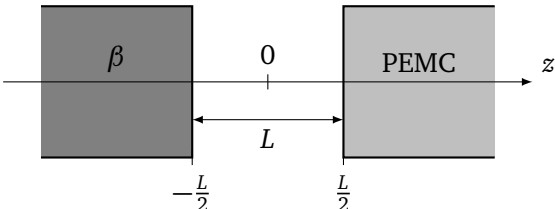

Figure 5:  Two semi-infinite slabs, a WSM parametrized by $\beta$ and a perfect electric conductor, with a vacuum gap of width $L$ in between.

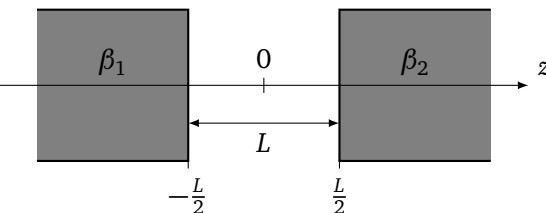

Figure 6:  Two semi-infinite slabs of WSMs, parametrized by $\beta_1$ and $\beta_2$, with a vacuum gap of width $L$ in between.

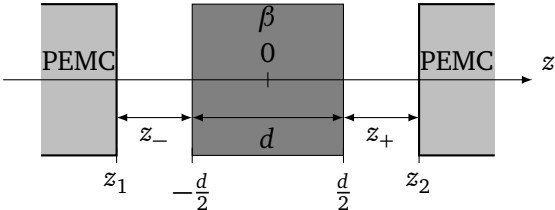

Figure 7:  A static slab of WSM with width $\ell$, and two semi-infinite PEC plates on either side.

## B  Polarization basis

We can define an orthonormal polarization basis $E_\mu^r$, i.e. with $E_\mu^{r\dagger} E_\mu^s = \delta_{rs}$, in which $\varepsilon_{\mu\nu k3} k_k$ is diagonal. The $r = 0, 3$ polarizations are analogous to longitudinal and timelike polarizations

$$E_i^0 = \frac{k_i}{|\mathbf{k}|}, \quad E_3^0 = 0, \quad E_i^3 = 0, \quad E_3^3 = 1, \tag{B.1}$$

where $|\mathbf{k}| = \sqrt{k_i k_i}$, while $r = 1, 2$ vectors are transversal

$$E_\mu^1 = \frac{1}{\sqrt{2}}\left(\widetilde{E}_\mu^1 - i\widetilde{E}_\mu^2\right), \qquad E_\mu^2 = \frac{1}{\sqrt{2}}\left(\widetilde{E}_\mu^1 + i\widetilde{E}_\mu^2\right), \tag{B.2}$$

and are built out of real vectors that obey

$$\widetilde{E}_i^2 = \varepsilon_{ijk} \frac{k_k}{|\mathbf{k}|} \widetilde{E}_j^1, \qquad \widetilde{E}_i^1 k_i = \widetilde{E}_i^2 k_i = 0, \tag{B.3}$$

which do not have to be defined explicitly for our purposes. A vector $v_\mu$ then decomposes as $v_\mu = E_\mu^r v_r$.

## C  Overview of regularized Casimir energy density expressions

| Setup | Energy density |
|---|---|
| PEMC – WSM | $\mathcal{E}_1 = \mathrm{Re} \int \dfrac{\mathrm{d}^3\mathbf{k}}{(2\pi)^3} \log\left(1 - \dfrac{k_c - |\mathbf{k}|}{k_c + |\mathbf{k}|} e^{-2|\mathbf{k}|L + 2i\alpha}\right)$ |
| PEMC – finite width WSM | $\mathcal{E}_2 = \mathrm{Re} \int \dfrac{\mathrm{d}^3\mathbf{k}}{(2\pi)^3} \log\left(1 - C_1 e^{-2|\mathbf{k}|L + 2i\alpha}\right)$ |
| WSM – WSM | $\mathcal{E}_3 = \mathrm{Re} \int \dfrac{\mathrm{d}^3\mathbf{k}}{(2\pi)^3} \log\left(1 - \dfrac{k_{c1}k_{c2} - |\mathbf{k}|(k_{c1} + k_{c2}) + |\mathbf{k}|^2}{k_{c1}k_{c2} + |\mathbf{k}|(k_{c1} + k_{c2}) + |\mathbf{k}|^2} e^{-2|\mathbf{k}|L}\right)$ |
| PEMC – WSM – PEMC | $\mathcal{E}_4 = \mathrm{Re} \int \dfrac{\mathrm{d}^3\mathbf{k}}{(2\pi)^3} \log\left(1 - C_1\left(e^{-2|\mathbf{k}|z_+ + 2i\alpha_2} + e^{2|\mathbf{k}|z_- - 2i\alpha_1}\right) - C_2 e^{-2|\mathbf{k}|(z_+ - z_-) + 2i(\alpha_2 - \alpha_1)}\right)$ |

where

$$C_1 = \frac{2(k_c^2 - |\mathbf{k}|^2)\sinh(k_c d)}{(|\mathbf{k}| + k_c)^2 e^{k_c d} - (|\mathbf{k}| - k_c)^2 e^{-k_c d}}, \qquad C_2 = 4\frac{(|\mathbf{k}|^2 + k_c^2)^2 - (k_c^2 - |\mathbf{k}|^2)^2 \cosh^2(k_c d)}{\left[(|\mathbf{k}| + k_c)^2 e^{k_c d} - (|\mathbf{k}| - k_c)^2 e^{-k_c d}\right]^2}. \tag{C.1}$$

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
