# Peer review of "Interplay between chiral media and perfect electromagnetic conductor plates: repulsive vs. attractive Casimir force transitions"

_SciPost Physics, doi:SciPost Phys. 15, 213 (2023)_

## Round 3 · Referee Report · Anonymous (Referee 1) · 2023-7-9

Report
The paper under the review studies the Casimir repulsion in systems consisting of perfect electromagnetic conducting plates and Weyl semimetal slabs. Similar problems have been addressed in many publications but for simpler geometries. The authors made several interesting predictions with regard to the amplitude, the sign, and the distance dependence of the Casimir force. Given the current interest in the Casimir repulsion and to Weyl semimetals, this is a relevant topic. However, before the manuscript can be recommended for publication the authors have to address the points listed below.
-
A few lines below Eq. (1) the authors claim that a constant part of $\theta$ is irrelevant for the equations of motion. This is not quite true. To derive equations of motion one has to integrate by parts in (1) which induces a surface Chern--Simons type contribution which is proportional to the discontinuity of $\theta$. Such terms affect the propagation of electromagnetic waves through interface and thus influence the Casimir force. The authors have to explain this point.
-
The authors use the same letter $\theta$ to denote the axion field in (1) and a parameter in the boundary conditions (3). This is confusing and has to be corrected.
-
What are the realistic physical values of the parameters used in this work? My calculations done on the back of an envelope show that the product of the typical value of $\beta$ for Weyl semimetals and the typical scale of distances in Casimir experiments is about 0.01 or smaller. The values of $\beta d$ and $\beta L$ on Figures 1-4 are much larger. Also, usual metals are much closer to perfect electric conductors than to perfect magnetic ones corresponding to $\theta'\simeq \pm \pi/2$. Thus, perhaps naively, one can suspect that the physically relevant regions on the figures are very small (practically a point on each figure). Do the authors know any realistic material which allows access other regions of the parameters?

---

## Round 3 · Referee Report · Anonymous (Referee 2) · 2023-8-17

Report
In this paper the authors develop a general method for computing the Casimir force between materials with varying electromagnetic properties in a parallel-slab geometry. The types of materials considered include perfect electromagnetic conductors (PEMCs), of which Weyl semimetals (WSMs) are a special case. By considering pairings of different types of materials and the use of intermediate layers the authors identify several scenarios where their calculations yield a repulsive force.
The search for a repulsive Casimir effect resulting from exotic electromagnetic properties is not a new endeavor, as the authors acknowledge, but to my knowledge several of the configurations considered here are novel. I believe the work presented is a useful, if perhaps incremental, addition to this line of inquiry, though I have several comments and questions before I recommend publication.
1) The use of $\theta$ needs to be amended, as it is currently refers to two distinct quantities.
2) The authors have cited several references that examine the Casimir effect in PEMCs and Weyl semimetals, including some already identifying repulsive effects. How do the results obtained here compare to already known results when directly comparable, e.g. the case of two WSMs in vacuum considered in Ref 14 and by Rong, et al. in Chin. Phys. Lett. 38, 084501?
3) Can you comment on how realistic the parameters necessary for the repulsive regimes identified here are? Specifically, is it reasonable to suppose that PEMCs with arbitrary $\theta'$, or really with any value in the repulsive regime, can actually be realized?
4) In real systems, materials can only rarely be well approximated with idealized boundary conditions. For instance, it is not difficult to calculate corrections to the ideal Casimir effect between conducting plates due to imperfect EM reflection coefficients/finite conductivity. Can the authors comment or speculate on what effect deviations from the idealized behavior considered here might have on the repulsive effects they've identified? Could the imperfect case introduce a force component that washes out the repulsive effect?
5) It would be better to have the diagrams of the geometries being considered included in the relevant sections instead of in a separate appendix.

---

## Round 6 · Referee Report · Anonymous (Referee 1) · 2023-10-13

Report

I am satisfied with the revision. I find this paper interesting and well written. I believe that it makes an important contribution to the field.

---

## Round 6 · Referee Report · Anonymous (Referee 2) · 2023-10-30

Report

I am satisfied with the author's responses and changes to the manuscript. The paper is well written and addresses interesting questions of current interest. I recommend publication without any additional changes.

---

## Round 6 · Author Response

Dear Editor,

we thank both Reviewers for their helpful feedback towards improving our manuscript.

Here follows our reply to their queries:

  • We have traded the $\theta$'s of the boundary conditions for $\cot\alpha$'s to avoid confusion and improve our plots.

  • On p5, we have added the sentence "The results of this section agree with [14, 15]" to make proper reference to related works [14,15].

  • To justify our plotting interval [0,20], we added the paragraph "Our β parameter is a rescaled version of the one typically found in the literature ... [27,33,34]." on page 3, which suggests that this interval is indeed physically relevant and that our results might go beyond pure theoretical interest.

  • Just before eq.(2), we commented with "In principle... work." about the possibility to have discontinuities in the $\theta(z)$ profile, related to Hall surface currents, see also the Green's function comment at the end of Section II.

  • We have decided to leave the different "plate geometry figures" gathered together in Appendix A to maintain the reading flow of Section III, where we want to attract most attention to our results/the Casimir force figures. If the Reviewer insists, we are willing to redistribute the various "plate geometry figures" over the associated Subsections of Section III.

In addition, we have also implemented some further improvements:

  • We added the references 23, 26 and 32-36

  • The comparison with the QED result in eqs.(20) and (23) is made more concrete using a proper next to leading order series expansion.

  • We have made slight language improvements in the title, abstract and main text wherever appropriate.

In summary, we hope that the updated version of our manuscript will be taken into account for possible publication in SciPost Phys. If any further Reviewer comments would arise, we are happy to continue the discussion.

Thank you for your attention.

Sincerely yours, Thomas Oosthuyse and David Dudal

---

## Round 6 · List of Changes

• All appearances of $\theta$ and $\theta^\prime$ have been rewritten in terms of $\alpha$.

  • On p5, we have added the sentence "The results of this section agree with [14, 15]" to make proper reference to related works [14,15].

  • We added the paragraph "Our $\beta$ parameter is a rescaled version of the one typically found in the literature ... [27,33,34]." on page 3.

  • Just before eq.(2), we commented with "In principle... work." about the possibility to have discontinuities in the $\theta(z)$ profile.

  • We made a comment on the Green's function related to Hall currents at the end of Section II.

  • The added references are 23, 26 and 32-36

  • In eqs.(20) and (23) we expanded to higher order.

  • We have made slight language improvements in the title, abstract and main text wherever appropriate.

---

## Editorial Decision

published